# Direct Comparison of Two Different Definitions with Biochemical Recurrence after Low-Dose-Rate Brachytherapy for Prostate Cancer

**Shinichi Takeuchi** [1], **Koji Iinuma** [1,*], **Keita Nakane** [1], **Masahiro Nakano** [2], **Makoto Kawase** [1], **Kota Kawase** [1], **Manabu Takai** [1], **Daiki Kato** [1], **Takayuki Mori** [3], **Hirota Takano** [3], **Tomoyasu Kumano** [3], **Masayuki Matsuo** [3] and **Takuya Koie** [1]

1 Department of Urology, Gifu University Graduate School of Medicine, Gifu 5011194, Japan
2 Department of Urology, Gifu Prefectural General Medical Center, Gifu 5008717, Japan
3 Department of Radiology, Gifu University Graduate School of Medicine, Gifu 5011194, Japan
* Correspondence: kiinuma@gifu-u.ac.jp; Tel.: +81-582306000

**Abstract:** We aimed to determine whether biochemical recurrence-free survival (BRFS) of patients with prostate cancer (PCa) who received low-dose-rate brachytherapy (LDR-BT) differed according to the definition of biochemical recurrence (BCR) after radical prostatectomy (RP) and the definition given by the Japanese Prostate Cancer Outcome Study of Permanent Iodine-125 Seed Implantation (J-POPS). We reviewed the clinical records of 476 consecutive patients with PCa who received LDR-BT at the Gifu University Hospital. The primary endpoint of this study was the difference in BRFS between the two aforementioned definitions. When the follow-up period ended, 74 (15.5%) and 20 (4.2%) patients had BCR according to the RP and J-POPS definitions, respectively. The 5-year BRFS rates were 85.0% and 96.9% for the RP and J-POPS definitions, respectively ($p < 0.005$). According to the RP definition, the 5-year BRFS rates were 80.6% in the group aged <63 years and 86.6% in those aged ≥63 years ($p = 0.050$). According to the J-POPS definition, the 5-year BRFS rates were 94.1% and 97.8% in the groups aged <63 years and ≥63 years, respectively ($p = 0.005$). The definition of recurrence in LDR-BT may need to be reconsidered.

**Keywords:** prostate cancer; low-dose-rate brachytherapy; biochemical recurrence; definition of biochemical recurrence

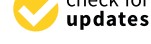

## 1. Introduction

Several guidelines recommend definitive local primary treatments for localized or selected locally advanced prostate cancer (PCa), including active surveillance, radical prostatectomy (RP) with pelvic lymphadenectomy, radiation therapy (RT), and androgen deprivation therapy (ADT) [1,2]. Low-dose-rate brachytherapy (LDR-BT) employs the use of a permanent radioactive seed embedded in the prostate and is one of the treatment options for patients diagnosed with low or good intermediate risk disease (low volume and Gleason grade [GG]2) [2,3], prostate volume (PV) less than 50 cm$^3$, and the International Prostate Symptom Score of 12 or less [4]. Recently, trimodality therapy combining ADT, external-beam RT (EBRT), and LDR-BT has become one of the standard treatments for high-risk and locally advanced PCa, and relatively better oncologic outcomes, including biochemical recurrence-free survival (BRFS), metastatic-free survival, and overall survival, have been achieved in these patients [5–7].

At present, RP and LDR-BT are widely recognized as curative treatments for localized PCa; however, obtaining data that compare oncological outcomes and quality of life of patients between the two treatment modalities is difficult despite the fact that patients themselves should be provided with data to help them select their treatment modality [8]. Moreover, conducting a prospective randomized controlled trial comparing LDR-BT and

RP [9] is fundamentally difficult, possibly in part because they use different definitions of biochemical recurrence (BCR) for each treatment, thus making directly comparing oncological outcomes impossible [10,11]. Although a prostate-specific antigen (PSA) level of >0.2 ng/mL is the most commonly defined BCR after RP (RP definition) [12], the Phoenix definition of 2 ng/mL from the post-treatment nadir point is commonly used for patients treated with RT, with a difference of approximately 10-fold between these two definitions [10]. Compared with the Phoenix definition, the definition according to the surgical threshold showed increased BCR by ~2% at 5 years and ~5% at 10 years and also a significantly increased BCR after dose-escalated EBRT; however, no difference was observed after LDR-BT [13,14]. Nevertheless, under Phoenix's definition, a clear distinction between benign PSA bounce and true BCR could not be determined [15]. Therefore, this study employed the definition of the nationwide, prospective Japanese Prostate Cancer Outcomes Study (J-POPS) of BCR after RT, in which a PSA of >1.0 ng/mL for at least three measurements is considered to identify BCR in a patient [15].

We aimed to determine whether the BRFS of patients with PCa who received LDR-BT differed according to different definitions of BCR after RP and as per the J-POPS.

## 2. Materials and Methods

### 2.1. Patient Population

Institutional Review Board of Gifu University (approval number: 29-106) has authorized this study. Since this study is a retrospective design, informed consent is not required. Since the results of retrospective and observational studies with existing materials and other data have previously been publicized, written consent was not necessary in accordance with the Japanese Ethics Committee and Ethical Guidelines. More information on this study can be viewed at https://rinri.med.gifu-u.ac.jp/esct/Common/document.aspx?ID=2911&VERSION=0&DOC_TYPE=210&PDF=1 (accessed on 24 December 2019).

We reviewed the clinical records of 476 consecutive patients with PCa who received LDR-BT at the Gifu University Hospital in Japan between August 2004 and August 2019. The enrolled patients had clinically organ-compromised or locally advanced PCa with no lymph node or distant metastases, based on the American Joint Committee on Cancer 8th edition of the Cancer Staging Manual [16]. All enrolled patients were categorized into risk groups with respect to the classification modality proposed by D'Amico [17]. The following clinical data were collected from the enrolled patients: age, initial serum PSA level, clinical T stage, biopsy Gleason grade (bGG) [3], the National Comprehensive Cancer Network (NCCN) risk classification [17], PV, presence or absence of ADT, and follow-up duration. Patients with a history of transurethral resection of the prostate or a uroflowmetry test result of <10 mL/s were not indicated for treatment with LDR-BT [6,7]. Since April 2010, all patients who had not undergone a colonoscopy within the past 2 years were evaluated for a complete colonoscopy prior to LDR-BT [18].

### 2.2. Treatment

Patients who had low-risk PCa or a PV of >50 mL had received neoadjuvant ADT at least 3 months prior to LDR-BT. Patients diagnosed with intermediate-risk PCa were administered ADT for nine months, which was followed by a combination of LDR-BT and/or EBRT. Patients who had high-risk PCa were given LDR-BT combined with EBRT and ADT for 24 months. Patients were implanted with loose 125I radioactive seeds (Oncoseed, Nippon Medi-Physics, Tokyo, Japan) by the Mick Applicator (Mick Radio-Nuclear Instruments, Bronx, NY, USA) or the ProLink® (Cincinnati, OH, USA) delivery system (C. R. Bard, Inc., Murray Hill, NJ, USA) under real-time confirmation by transrectal ultrasound transperineally into the prostate [19]. A minimum peripheral dose of 145 Gy was prescribed for LDR-BT alone, and 104 Gy for the combination of LDR-BT and EBRT. When EBRT was combined, a total of 40 Gy in 2-Gy fractions was irradiated to the prostate and seminal vesicles within 1 month after LDR-BT. In all cases, a modified peripheral loading technique was used after pre-planning for seed implantation [20].

### 2.3. Post-Dosimetric Evaluation

Treatment design and post-implant dosimetric evaluation were carried out in accordance with the latest American Medical Association Task Group 43 protocols and Variseed version 7.1 (Varian Medical Systems, Palo Alto, CA, USA). For post-implantation dose measurements, computed tomography (CT) and magnetic resonance imaging (MRI) were carried out 1 month from LDR-BT. A CT with a 16- or 64-detector array CT scanner (Light-Speed Ultra 16/Discovery CT 750 HD; GE Healthcare, Milwaukee, WI, USA) was used [21]. Also, an MRI using a 5-channel SENSE cardiac coil was performed in easy-breathing conditions using a slice thickness of 3 mm and no cross-gap (Intera Achieva 1.5 T/Intra Achieva Nova Dual 1.5 T Pulsar: Philips Medical Systems, Philips Medical Systems, Eindhoven, The Netherlands) [21]. In this study, the following dosimetric parameters were evaluated: the minimum percentage of the prostate gland received at 90% (D90), the percentage of the PV receiving 100% of the specified minimum peripheral dose (V100), the percentage of the rectal volume receiving 100% of the specified dose (RV100), and the biologically effective dose (BED).

### 2.4. Follow-Up Schedule

A follow-up for all of the patients was conducted at 3–6-month intervals for 5 years and then at 6–12-month intervals thereafter. At the follow-up, their history was obtained, a physical examination was performed, and PSA was assessed; testosterone levels were also measured in patients who received ADT. The follow-up period was from the end of RT to the last follow-up date or the date of death; the RP and J-POPS definitions were adopted to identify BCR following LDR-BT [12,15]. A temporary rise in the PSA level indicated a PSA bounce and was not considered BCR.

### 2.5. Statistical Analysis

The primary endpoint of this study was the difference in BRFS between the two aforementioned definitions. The secondary endpoint of the study comprised evaluating BRFS and clinicopathological covariates with respect to BCR. For data analysis, we used JMP 14 (SAS Institute Inc., Cary, NC, USA). BRFS after LDR-BT was analyzed using the Kaplan–Meier method. A subgroup analysis for BRFS was performed using the log-rank test. As the cutoff value using the area under the receiver operating characteristic curve differed according to the two definitions of BCR, the median value was used as the cutoff variable for the covariates in this study. Two-tailed $p$-values $< 0.05$ were considered statistically significant in all of the cases.

## 3. Results

### 3.1. Patient Characteristics

Table 1 presents the characteristics of the 476 patients who met the study criteria and were enrolled in the study.

**Table 1.** Patient characteristics.

| | |
|---|---|
| Age (year, median, IQR) | 66 (50–81) |
| PSA (ng/mL, median, IQR) | 6.44 (1.7–60.8) |
| Clinical T stage (number, %) | |
| T1c | 252 (52.9) |
| T2a | 136 (28.6) |
| T2b | 29 (6.1) |
| T2c | 47 (9.9) |
| T3a | 11 (2.3) |
| T3b | 1 (0.2) |
| Gleason Group Grade (number, %) | 2 (1–5) |
| 1 | 199 (41.8) |
| 2 | 166 (34.9) |
| 3 | 67 (14.1) |

**Table 1.** *Cont.*

| | |
|---|---|
| 4 | 28 (5.9) |
| 5 | 16 (3.3) |
| NCCN risk classification (number, %) | |
| Low | 169 (35.5) |
| Intermediate | 248 (52.1) |
| High | 59 (12.4) |
| Prostate volume at LDR-BT (mL, median, IQR) | 23.4 (13.8–53.0) |
| Neoadjuvant ADT (number, %) | 369 (77.5) |
| Follow-up period (month, median, IQR) | 84 (1–216) |

IQR, interquartile range; PSA, prostate-specific antigen; NCCN, National Comprehensive Cancer Network; LDR-BT, Iodine-125 low-dose-rate brachytherapy; ADT, androgen deprivation therapy.

Among the enrolled patients, the median D90 was 119.6% (interquartile range [IQR], 111.3–127.0%), V100 was 96.6% (IQR, 94.4–97.9%), and BED was 193.8 Gy (IQR, 177.9–209.7 Gy). The median BED was 183.2 Gy for the LDR-BT group and 210.1 Gy for the LDR-BT + EBRT group. The anatomic data from the enrolled patients indicated a median RV100 of 0.29 mL (IQR, 0.07–0.73 mL). The median RV100 was 0.31 and 0.27 mL in the LDR-BT and LDR-BT + EBRT groups, respectively.

*3.2. Oncological Outcomes*

In the follow-up period, 74 (15.5%) and 20 (4.2%) patients had BCR based on the RP and J-POPS definitions. Although no deaths from PCa were recorded, deaths from other causes were observed in 16 patients. The etiologies of the patients who died of other causes were malignant neoplasms of other organs in seven, cardiac disease in three, infectious diseases in two, and cerebral infarction, traffic accidents, interstitial pneumonia, and unknown cause of death in one patient each. Regarding the RP definition, the 3-, 5-, and 10-year BRFS rates were 86.7%, 85.0%, and 83.9%, respectively (Figure 1). When BRFS was examined using the J-POPS definition, the 3-, 5-, and 10-year BRFS rates were 97.9%, 96.9%, and 94.8%, respectively (Figure 1). BRFS rate based on the RP definition was significantly worse than that based on the J-POPS definition (*p* < 0.005).

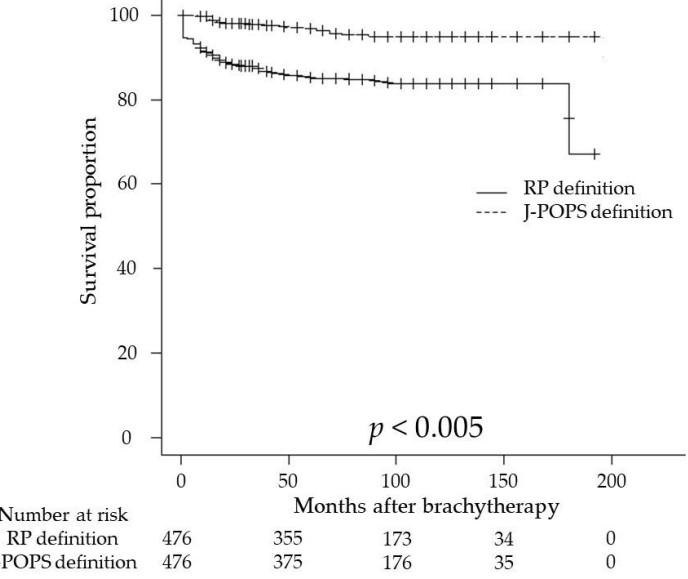

**Figure 1.** Kaplan–Meier estimates of biochemical recurrence-free survival (BRFS) in patients with prostate cancer who received low-dose-rate brachytherapy (LDR-BT) were calculated according to two definitions of biochemical recurrence (BCR) as follows: the RP definition as a prostate-specific antigen (PSA) of ≥0.2 ng/mL after LDR-BT and the J-POPS definition as PSA of >1.0 ng/mL for at least three measurements after LDR-BT. The 5-year BRFS rate was 85.0% for the RP definition and 96.9% for the J-POPS definition (*p* < 0.005).

BCR in the two groups with a median age of 63 years was compared using the RP and J-POPS definitions (Figure 2). With the RP definition, the BRFS rates were 83.2% at 3 years, 80.6% at 5 years, and 78.3% at 10 years in the group aged < 63 years. For patients aged ≥ 63 years, the 3-, 5-, and 10-year BRFS rates were 87.9%, 86.6%, and 86.0%, respectively (Figure 2A). Although no significant difference was observed, the BCR rate tended to be higher in the population aged < 63 years than in those aged ≥ 63 years ($p = 0.050$; Figure 2A). Using the J-POPS definition, the BRFS rates were 95.1% at 3 years, 94.1% at 5 years, and 89.7% at 10 years in the group aged < 63 years. For patients aged ≥ 63 years, the 3-, 5-, and 10-year BRFS rates were 98.8%, 97.8%, and 96.8%, respectively (Figure 2B). The BCR rate was significantly higher in the population aged < 63 years than in those aged ≥ 63 years ($p = 0.005$; Figure 2B).

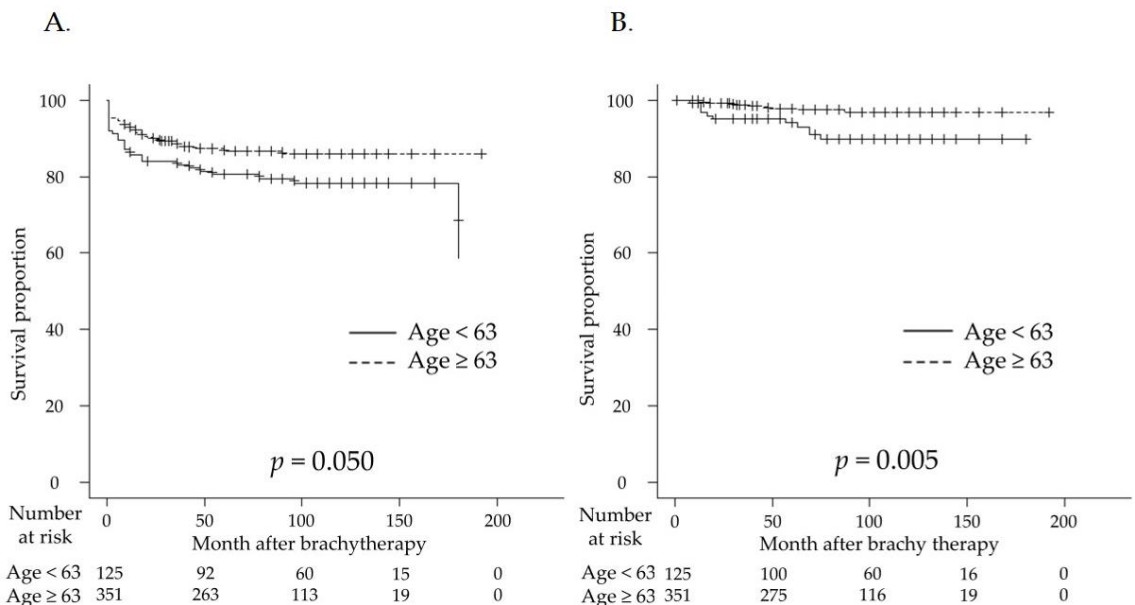

**Figure 2.** Kaplan–Meier estimates of biochemical recurrence-free survival (BRFS) in patients with prostate cancer who received low-dose-rate brachytherapy were calculated according to patient age. (**A**) With the RP definition, the 5-year BRFS rates were 80.6% in the group aged <63 years and 86.6% in those aged ≥63 years. The biochemical recurrence rate (BCR) tended to be higher in the population aged <63 years than in those aged ≥63 years ($p = 0.050$). (**B**) With the J-POPS definition, the 5-year BRFS rates in the groups aged <63 years and ≥63 years were 94.1% and 97.8%, respectively. The BCR rate was significantly higher in the population aged <63 years than in those aged ≥63 years ($p = 0.005$).

Other factors, including the initial PSA level, clinical T stage, bGG, NCCN risk stratification, and presence of absence of adjuvant ADT administration, did not differ significantly either with the RP definition or the J-POPS definition (Figures S1–S10).

## 4. Discussion

Although LDR-BT is primarily performed in low- and intermediate-risk PCa, it is increasingly being conducted to boost EBRT and/or ADT for dose escalation in unfavorable intermediate- and high-risk PCa [1]. Andrea et al. [22] investigated and reported whether PSA was a predictor of disease-free survival at 4–5 years in patients undergoing LDR-BT. Of the 1434 patients with PCa who received LDR-BT monotherapy, 63 (4.4%) developed BCR according to Phoenix's definition [22]. Among them, only one case (0.7%) of recurrent PCa without BCR was noted [22]. Of the remaining 62 cases, the site of recurrence was determined in 21 cases (34%), of which 8 had local recurrence and 13 had distant metastasis [22]. The remaining 41 patients (66%) had only BCR and no local recurrence or distant metastasis [22]. The first report from the multicenter randomized Androgen Suppression

Combined with Elective Nodal and Dose Escalated Radiation (ASCENDE-RT) trial examined the comparative outcomes of dose-escalated EBRT (DE-EBRT) versus LDR-BT [23]. The 5-, 7-, and 9-year BRFS rates according to the Phoenix definition were 84%, 75%, and 62% in the DE-EBRT group and 89%, 86%, and 83% in the LDR-BT group, respectively ($p < 0.001$) [23]. LDR-BT was significantly associated with improved BRFS in patients with both intermediate- and high-risk PCa ($p = 0.003$ and $p = 0.048$, respectively) [23]. In the multivariate analysis, LDR-BT, the percent positive biopsy core, clinical T stage, and initial PSA level were significantly correlated with BCR [23]. Similarly, regarding overall survival (OS), LDR-BT had a relatively better outcome, with age as the only significant prognostic factor in the multivariate analysis [23]. Of all of the enrolled patients, 76 (19.1%) developed BCR and 35 (8.8%) had distant metastases; metastasis-free survival was similar in both groups, whereas the percent positive biopsy core, clinical T stage, and GG were significant independent predictors of metastasis in the multivariate analysis [23]. In the ASCENDE-RT trial, which recently evaluated 568 patients with PCa undergoing LDR-BT and EBRT, 79 (13.9%) had BCR according to the Phoenix definition, with an estimated BRFS rate of 84% and a median follow-up of 4.5 years (IQR, 3.2–5.8 years) [5]. Conversely, with respect to the total dose, increasing the BED in 10-Gy increments from 140 Gy to 200 Gy improved the BRFS rate and was associated with the upward slope of the dose–response sigmoid curve observed at other disease sites [24]. For PCa with any risk, increasing the BED to ≥200 Gy was not associated with an improvement in the 5-year BCR [24]. Regarding the slope coefficient of the meta-regression, each 10-Gy increase in the BED did not significantly ($p > 0.05$) improve BRFS, by almost zero units, which is consistent with a plateau in the dose–response sigmoid curve [25]. Previous reports have indicated LDR-BT as an effective treatment for PCa, although its therapeutic effect remains almost the same after exceeding the prescribed dose. Meanwhile, as developing local/regional recurrence, metastasis, and PCa-related death are observed in some patients with PCa, reconsidering not only the choice of treatment and the prescribed dose, but also the definition of recurrence, depending on the type of RT, may be necessary.

The American Society for Therapeutic Radiology and Oncology and Radiation Therapy Oncology Group defined BCR after RT as a nadir PSA value of +2 ng/mL, which has been widely adopted as the Phoenix definition [10]. As a rationale, the Phoenix definition is necessary to avoid a large number of "false positives", because EBRT commonly preserves PSA-secreting glands [14]. However, a fair evaluation of the two definitions for patients with PCa receiving RP or RT compels us to conclude that the RT definition leads to a large lead-time bias in favor of RT in the reporting of actuarial results [14]. As with RP, the high threshold of the Phoenix definition has also been questioned with LDR-BT for the prostate, since LDR-BT often results in the complete ablation of glandular tissue and PSA values remain undetectable [13,22,23]. Moreover, the Phoenix definition may significantly underestimate the BCR at a follow-up of up to 15 years, and there is the antinomy with respect to sensitivity and specificity for defining the BCR [25]. Based on a report from Mount Sinai Hospital, which included 2634 patients with clinical T1-T4N0M0 PCa, 293 (11.1%) met the Phoenix definition of BCR and 457 (17.5%) met the RP definition ($p < 0.001$) [25]. BRFS according to the Phoenix definition was superior to that according to the RP definition at 5 and 10 years, but not at 15 years [25]. For patients with PCa who did or did not receive ADT, the BCR rate was lower when using the Phoenix definition than when using the RP definition, which was 6.0% vs. 10.0% ($p < 0.001$) and 9.3% vs. 12.4% ($p < 0.001$), respectively [25]. In the multivariate analysis, BCR was statistically correlated with the initial PSA, clinical T stage, GG, and BED for both the Phoenix and RP definitions [25]. By contrast, the duration of ADT and patient age were significantly associated with the RP definition with respect to BCR but not with the Phoenix definition [25]. In their study, 64 patients (2.4%) died of PCa, with a median time from BCR to death of 3.7 years according to Phoenix's definition and 5.8 years according to the RP definition [25]. By contrast, 195 (8.4%) and 137 (5.9%) patients had BCR based on the Phoenix and J-POPS definitions, respectively, with a median follow-up time of 60.0 months [15]. Clinical recurrence, distant

metastasis, cause-specific death, and other causes of death were observed in 49 (2.1%), 41 (1.8%), 7 (0.3%), and 55 (2.4%) patients with PCa, respectively, who received LDR-RT [15]. The 5-year BRFS rates were 89.1% and 91.6%, as defined by the Phoenix and J-POPS, respectively [15]. Of the 84 patients who achieved BCR according to the Phoenix definition alone, 1.2% had clinical recurrence before BCR and only 7.1% received salvage treatment after BCR [15]. A spontaneous decrease in PSA levels was observed in 93.5% of patients after BCR [15]. Of the 22 patients who met the J-POPS definition, 18.2% received salvage treatment after BCR without clinical recurrence [15]. In the absence of salvage treatment, the spontaneous reduction of PSA levels was observed in only 22.2% of the patients [15]. Ito et al. [15] concluded that the J-POPS definition, compared with the Phoenix definition, could enable a clear delineation of treatment failure groups and may avoid inadequate local treatment for cases with undetectable metastases. The Phoenix definition may be useful for determining recurrence after RT, however, it may not be appropriate for the current situation, considering that it has been >15 years since this definition was proposed, and that there are various treatment methods for RT. Moreover, since ADT must be administered for several months during RT, the recovery of testosterone may not correlate with an increase in PSA. Furthermore, metastasis and death from PCa occur in more than ten percent of patients; therefore, the definition of recurrence after RT may need to be reevaluated in the future. Although two definitions of BCR were used and examined in this study, it was difficult to determine which definition was more useful for oncologic outcomes in patients who underwent LDR-BT. In order to develop a definition of BCR for LDR-BT, long-term follow-up with a homogeneous patient population and the identification of clinically useful variables seems to be necessary in the near future.

In the present study, age was a significant predictor of BCR, which is a finding similar to that in our previous report [7]. Previous studies have reported that age is associated with OS [23], some with BCR [5,7,15,25] and some with no statistically difference between age and recurrence [22,26], although no consensus has been reached thus far. The ASCENDE-RT trial demonstrated a significant correlation between BCR and increased all-cause mortality (hazard ratio [HR], 6.30; $p < 0.001$) and a significant reduction in BCR with LDR-BT boost (HR, 2.04; $p = 0.004$) [23]. Although a longer follow-up may indicate OS benefits of LDR-BT, this may be debatable [23]. The ASCENDE-RT trial registered 398 patients with a median age of 68 years [23]. The relatively small number of elderly patients and the longer interval between local recurrence and life-threatening disease suggest that the potential survival benefit associated with improved local control may be compromised by potential competing causes of mortality [23]. Therefore, using patient age as a predictor of BCR and OS remains controversial.

This study had several limitations. First, because this was a retrospective study with data from a single center, selective biases in determining definitive therapy for PCa, such as the preference of attending physicians and patients, may have been present. Second, when using the RP definition, we did not define a duration of time for PSA to decrease to <0.2 ng/mL after LDR-BT; PSA in patients with PCa who received only EBRT or LDR-BT may require more time to reach <0.2 ng/mL. Therefore, establishing a criterion for the time required for PSA to decrease may be necessary. Third, we were unable to demonstrate whether the difference in BRFS between the two definitions used in this study affected the OS or cancer-specific survival. Lastly, we did not examine the incidence of secondary cancers. Therefore, a detailed prognostic study is required.

## 5. Conclusions

Patients who met the RP definition had a significantly higher BCR than those who met the J-POPS definition. In patients who received LDR-BT for PCa, younger patients tended to have a higher BCR than older patients. Therefore, the definition of recurrence in LDR-BT can be reconsidered. Additionally, LDR-BT for PCa in younger patients may require clear criteria for treatment selection and careful follow-up.

**Supplementary Materials:** The following supporting information can be downloaded at: https://www.mdpi.com/article/10.3390/curroncol30030212/s1, Figures S1–S10: Kaplan-Meier estimates of biochemical recurrence-free survival (BRFS) for patients with PCa receiving low-dose-rate brachytherapy were calculated using the definition of biochemical recurrence after radical prostatectomy (RP) and the definition given in the Japanese Prostate Cancer Outcomes Study (J-POPS) with respect to pretreatment clinical covariates, including initial prostate-specific antigen (PSA) level, clinical T stage, biopsy Gleason Grade Group (bGG), National Comprehensive Cancer Network (NCCN) risk classification, and adjuvant androgen deprivation therapy (ADT).

**Author Contributions:** Conceptualization, S.T., K.I. and T.K. (Takuya Koie); methodology, S.T. and K.I.; investigation, S.T. and K.I.; resources, S.T., K.I., K.N., M.N., M.K., K.K., M.T., D.K., T.M., H.T. and T.K. (Tomoyasu Kumano); data curation, S.T. and K.I.; supervision, M.M.; writing—original draft preparation, S.T.; writing-review and editing, K.I. All authors have read and agreed to the published version of the manuscript.

**Funding:** This study received no external funding.

**Institutional Review Board Statement:** The protocol for this study was approved by the Institutional Review Board of Gifu University (number: 29-106) on 5 July 2017. All procedures performed in studies involving human participants were in accordance with the ethical standards of the institutional and/or national research committee and with the 1964 Helsinki declaration and its later amendments or comparable ethical standards.

**Informed Consent Statement:** For this type of study formal consent is not required. Pursuant to the provisions of the ethics committee and the ethic guideline in Japan, written consent was not required in exchange for public disclosure of study information in the case of retrospective and/or observational study using a material such as the existing documentation. The study information was open for the public consumption at https://www.med.gifu-u.ac.jp/visitors/disclosure/docs/2020-210.pdf (accessed on 30 September 2022).

**Data Availability Statement:** The data presented in this study are available on request from the corresponding author. The data are not publicly available due to privacy and ethical reasons.

**Conflicts of Interest:** The authors declare no conflict of interest.

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
