# Peer review of "Direct Comparison of Two Different Definitions with Biochemical Recurrence after Low-Dose-Rate Brachytherapy for Prostate Cancer"

_curroncol, doi:10.3390/curroncol30030212_

Round 1

Reviewer 1 Report

This is an interesting and well-written manuscript. 

There are some point to clarify.

Title please avoid using acronism (J-POPS)

Introduction

line 42. LD-BCR is one of the options, is n

line 61 is it a low or high sensitivity? what about the JPOP sensitivity? I would

delete this sentence.

Figure 1

5r-y BCRFS is 94 % and 97%? In the graph and text are 85 and 96%. 

 Line 196-199 Please include this analysys in supplementary materials

Discussion

line 218 in reference 24, which definition of BCR was used?

line 222  "of 2936 enrolled patients". Please delete enrolled 

please introduce the study reference 23.

Using both definitiion recurrence rate should vary. The question I have is  which one is more beneficial for the patients and why?

Conclusion instead of repeting the age, just change it for younger/older patients

The state of the definition of LDR-BT should be reconsider is too strong for a study with the limitations mentioned. Please soften it.

Author Response

The authors appreciate the academic editor’s comments. The authors’ point-by-point responses to the comments are given in attached file.

Reviewer 2 Report

1.       The authors must be avoided using abbreviations in the title.

2.       The discussion section is too long and confusing. It is hard to read and understand.

3.       Table 2 is unnecessary.

4.       I think comparing two different threshold is not the way to suggest a new threshold. This might be performed with having a long-term follow-up in a homogeneous patient group and defining a new threshold with accompanying clinical variables.

Author Response

(The authors gave the same response as above.)

Round 2

Reviewer 2 Report

I would like to thank to the authors for their efforts to revise this manuscript. However, the revisions did not fulfill my concerns related to study design.